# Secreted Factors from Stem Cells of Human Exfoliated Deciduous Teeth Directly Activate Endothelial Cells to Promote All Processes of Angiogenesis

**DOI:** 10.3390/cells9112385

**Published:** 2020-10-31

**Authors:** Makoto Kato, Shin Tsunekawa, Nobuhisa Nakamura, Emiri Miura-Yura, Yuichiro Yamada, Yusuke Hayashi, Hiromi Nakai-Shimoda, Saeko Asano, Tomohide Hayami, Mikio Motegi, Emi Asano-Hayami, Sachiko Sasajima, Yoshiaki Morishita, Tatsuhito Himeno, Masaki Kondo, Yoshiro Kato, Takako Izumoto-Akita, Akihito Yamamoto, Keiko Naruse, Jiro Nakamura, Hideki Kamiya

**Affiliations:** 1Division of Diabetes, Department of Internal Medicine, Aichi Medical University School of Medicine, Nagakute 480-1195, Japan; katou.makoto.764@mail.aichi-med-u.ac.jp (M.K.); tsune87@aichi-med-u.ac.jp (S.T.); yura.emiri.036@mail.aichi-med-u.ac.jp (E.M.-Y.); yamada.yuuichirou.306@mail.aichi-med-u.ac.jp (Y.Y.); hayashi.yuusuke.268@mail.aichi-med-u.ac.jp (Y.H.); nakai.hiromi.706@mail.aichi-med-u.ac.jp (H.N.-S.); noda.saeko.707@mail.aichi-med-u.ac.jp (S.A.); hayami.tomohide.884@mail.aichi-med-u.ac.jp (T.H.); motegi.mikio.890@mail.aichi-med-u.ac.jp (M.M.); asano.emi.870@mail.aichi-med-u.ac.jp (E.A.-H.); sasajima.sachiko.016@mail.aichi-med-u.ac.jp (S.S.); morishita.yoshiaki.517@mail.aichi-med-u.ac.jp (Y.M.); himeno.tatsuhito.869@mail.aichi-med-u.ac.jp (T.H.); kondou.masaki.330@mail.aichi-med-u.ac.jp (M.K.); katou.yoshirou.524@mail.aichi-med-u.ac.jp (Y.K.); nakamura.jirou.574@mail.aichi-med-u.ac.jp (J.N.); 2Department of Internal Medicine, School of Dentistry, Aichi Gakuin University, Nagoya 464-8651, Japan; nnaka@dpc.agu.ac.jp (N.N.); narusek@dpc.agu.ac.jp (K.N.); 3Department of Oral and Maxillofacial Surgery, Nagoya University Graduate School of Medicine, Nagoya 466-8550, Japan; tan_chun_0531@yahoo.co.jp; 4Department of Histology and Oral Histology, Institute of Biomedical Sciences, Tokushima University Graduate School, Tokushima 770-8501, Japan; akihito@tokushima-u.ac.jp

**Keywords:** angiogenesis, endothelial cell, SHED, conditioned medium

## Abstract

Diabetes is a major risk factor for atherosclerosis and ischemic vascular diseases. Recently, regenerative medicine is expected to be a novel therapy for ischemic diseases. Our previous studies have reported that transplantation of stem cells promoted therapeutic angiogenesis for diabetic neuropathy and ischemic vascular disease in a paracrine manner, but the precise mechanism is unclear. Therefore, we examined whether secreted factors from stem cells had direct beneficial effects on endothelial cells to promote angiogenesis. The soluble factors were collected as conditioned medium (CM) 48 h after culturing stem cells from human exfoliated deciduous teeth (SHED) in serum-free DMEM. SHED-CM significantly increased cell viability of human umbilical vein endothelial cells (HUVECs) in MTT assays and accelerated HUVECs migration in wound healing and Boyden chamber assays. In a Matrigel plug assay of mice, the migrated number of primary endothelial cells was markedly increased in the plug containing SHED-CM or SHED suspension. SHED-CM induced complex tubular structures of HUVECs in a tube formation assay. Furthermore, SHED-CM significantly increased neovascularization from the primary rat aorta, indicating that SHED-CM stimulated primary endothelial cells to promote comprehensive angiogenesis processes. The angiogenic effects of SHED-CM were the same or greater than the effective concentration of VEGF. In conclusion, SHED-CM directly stimulates vascular endothelial cells to promote angiogenesis and is promising for future clinical application.

## 1. Introduction

Ischemic vascular diseases including cardiovascular disease (CVD) and peripheral artery disease (PAD) are based on arteriosclerosis and increasing worldwide. CVD remains the main cause of death, accounting for 16% of all deaths [1]. The morbidity and mortality associated with PAD are equal to or higher than those associated with CVD [2]. Diabetes is a major risk factor for atherosclerosis and patients with diabetes have approximately 2- to 4-fold greater risk of developing ischemic vascular diseases than non-diabetic patients. The death rate from ischemic vascular disease among diabetic patients in the United States and Japan has been decreasing, but it remains the leading cause of death in the United States [3] and the third in Japan [4]. In the field of therapeutic angiogenesis, many studies have been undertaken to identify angiogenic factors and develop new treatment strategies, such as gene therapy and regenerative medicine, for ischemic diseases [5,6]. Angiogenesis is a physiological process that promotes outgrowth of new blood vessels from pre-existing vasculature, which plays an important role in tissue growth and survival [7]. Transplantation of stem cells, especially mesenchymal stem cells (MSCs), is expected to be a new strategy to promote angiogenesis [8,9,10]. Basic and clinical research has suggested that the central mechanism of angiogenesis induced by cell transplantation is mediated through paracrine effects of soluble angiogenic factors secreted from transplanted stem cells [11] because the transplanted cells show poor differentiation and low survival rates.

Dental pulp stem cells are located in the perivascular niche of dental pulp and are thought to originate from the cranial neural crest, which express markers of both embryonic stem cells and MSCs [12]. From the viewpoint of phenotypes, including the proliferation rate, cell differentiation potency, and cell surface proteins, dental pulp stem cells from human exfoliated deciduous teeth (SHED) are usually distinct from dental pulp stem cells from permanent teeth (DPSCs) [13]. SHED have a high capacity for proliferation and differentiate into various of cell types such as neural cells, adipocytes, osteoblasts, and chondrocytes. SHED also express many genes encoding extracellular and cell surface proteins at the same levels or higher than those in human bone marrow-derived MSCs [12]. Furthermore, SHED can be easily collected from exfoliated deciduous teeth with minimal invasiveness and applied to autologous treatments. Based on these phenotypes and easy isolation, SHED have received considerable attention as a stem cell source for regenerative medicine.

Our previous studies and others have demonstrated that engrafted SHED in various animal models of diseases, including diabetes mellitus [14], spinal cord injury [15], systemic lupus erythematosus [16], and ischemic brain injury [17], promote significant recovery through endogenous tissue-repairing activities. Recently, we also reported that transplantation of BM-MSCs [18] or DPSCs [19] into limb skeletal muscles improves diabetic polyneuropathy (DPN) in diabetic animal models, but the transplanted cells show poor differentiation and low survival rates. These observations suggest that the improvement is attributable to paracrine effects and the secreted soluble factors from transplanted cells play a vital role [19]. Accordingly, our subsequent study showed that secreted factors from SHED, which can be collected as serum-free conditioned medium (CM), prevent deterioration of DPN in mice and only secreted factors of less than 6 kDa in SHED-CM promoted neurite outgrowth of mouse dorsal root ganglion (DRG) neurons [20]. Interestingly, both transplantation of stem cells and administration of SHED-CM improved blood flow in peripheral neurotrophic blood vessels and increased the capillary number to muscle fiber ratio in diabetic mice, which indicated that soluble factors from the stem cells promoted angiogenesis [20]. Furthermore, transplantation of SHED has been reported to up-regulate the level of serum VEGF and heal hindlimb ulcers in rat diabetic models [21]. Recent studies by us and others have reported that SHED secrete angiogenic trophic factors such as VEGF, HGF, FGF2, NGF, and BDNF [20,22,23,24], but the concentrations of these factors were not sufficient for bioactivity. In addition, we previously reported that conditioned medium from SHED contained enough exosomes to protectively effect on pancreatic beta-cells and MSC-derived exosomes have been reported their efficacy in animal models of myocardial infarction, stroke, and hind-limb ischemia [25,26,27,28,29].

These reports led us to believe that soluble factors from SHED might have a great potential to induce angiogenesis. However, it is unclear whether SHED-CM directly activates endothelial cells for angiogenesis. Therefore, in this study, we examined whether secreted factors in SHED-CM had beneficial effects on endothelial cells to promote angiogenesis.

## 2. Materials and Methods

### 2.1. Preparation of SHED-CM

Exfoliated human deciduous teeth were obtained from children aged 6–12 years and used in accordance with a protocol approved by the Nagoya University Ethics Committee (H-73, 2003). SHED were collected from dental pulp and cultured as described previously [12]. SHED passaged 10 to 14 times were used in experiments. To obtain SHED-CM, SHED were cultured to 80% confluence in 10-cm dishes, which was approximately 6 × 10^6^ cells. The cells were rinsed three times with phosphate-buffered saline (PBS) and then cultured in serum-free Dulbecco’s modified Eagle’s medium (DMEM) for 48 h at 37 °C. The media were collected and centrifuged at 3000× *g* for 5 min at 4 °C, and the supernatant was collected as SHED-CM as described previously [30].

### 2.2. Separation of SHED-CM According to Molecular Mass/kDa

Ultrafiltration devices were used to separate SHED-CM into low and high molecular mass fractions, SHED-CM was centrifuged at 5000× *g* for 1 h at 4 °C in Amicon^®^ Ultra-15 Centrifugal Filter Units (Millipore, Billerica, MA, USA). The filtrate containing the low molecular weight fraction (<6 kDa) and supernatant containing the high molecular weight fraction (>6 kDa) were diluted to the original volume with serum-free DMEM.

### 2.3. Isolation and Purification of Exosomes from SHED-CM

SHED at 80% confluence were rinsed three times with PBS and cultured for 48 h in serum-free DMEM. The media were collected and centrifuged at 3000× *g* for 5 min, followed by further centrifugation at 1500× *g* for 10 min at 4 °C. The supernatant was filtered through a 0.22-µm pore filter (Millipore) to remove whole cells and cellular debris. The CM was placed in a Thinwall Polypropylene Tube (Beckman Coulter, Brea, CA, USA) and ultracentrifuged at 100,000× *g* for 110 min at 4 °C (L-70; Beckman Coulter, Indianapolis, IN, USA). The pellet enriched with exosomes was resuspended in serum-free DMEM. The presence of exosomes was confirmed using transmission electron microscopy.

### 2.4. Human Umbilical Vein Endothelial Cells (HUVECs)

Human umbilical vein endothelial cells (HUVECs; Lonza Japan, Tokyo, Japan) were cultured in EGM™-2 BulletKit™ (Lonza Japan). Cells passaged four to seven times were used in angiogenesis assays.

### 2.5. Animals

Male C57BL/6J mice and Sprague-Dawley (SD) rats were obtained from Nihon SLC (Shizuoka, Japan). The mice and rats were housed in an aseptic animal room under controlled temperature (20–24 °C) in a 12-h light/dark cycle and allowed free access to standard laboratory chow and water. All experimental protocols were approved by the Department of Animal Experiments at Aichi Medical University (2019-112).

### 2.6. MTT Assay

HUVECs were seeded in 96-well plates at a density of 1 × 10^4^ cells per well in 100 µL EGM™-2 and cultured for 24 h at 37 °C. After rinsing the cells three times with PBS, the cells were cultured in 100 µL of six different media [DMEM, DMEM with 19.1 ng/mL VEGF (VEGF165; R&D Systems, Abingdon, UK), whole SHED-CM, the <6 kDa fraction, >6 kDa fraction, or DMEM with exosomes] for 24 or 48 h at 37 °C (*n* = 5). VEGF, which has an experimentally effective concentration from 10 to 20 ng/mL [28,29,30], is well known to promote angiogenesis [31]. Therefore, in this study, 19.1 ng/mL VEGF was applied as a positive control. Then, 10 µL 3-(4,5-dimethylthiazol-2-yl)-2,5-diphenyltetrazolium bromide (MTT; Dojindo Laboratories, Kumamoto, Japan) was added to each well at a final concentration of 0.5 mg/mL. After a further 4 h of incubation, the cells were lysed with 100 µL of 0.04 mol/L HCl in isopropanol. Cell viability was calculated by measuring absorbance at 570 nm with a microplate reader (SpectraMax M5; Molecular Devices; Sunnyvale, CA, USA).

### 2.7. Wound Healing Assay

HUVECs were cultured to confluency in glass-based dishes (IWAKI, Shizuoka, Japan) with EGM™-2. Using a sterile P-1000 pipette tip, the HUVEC monolayer was scratched linearly. After washing, the cells were incubated for 18 h at 37 °C in the six different culture media described above (*n* = 5). Images at the time of scratching were captured to measure the initial distance of wound at seven random sites with a charge-coupled device camera (DP70; Olympus Optical, Tokyo, Japan). Then, the distance between the leading edge covered by cells 18 h later were measured to evaluate the migration distance at same sites. In addition, the wound field was measured by ImageJ software (National Institutes of Health, Bethesda, MD, USA; http://rsbweb.nih.gov/ij/) at seven random sites at 0 and 18 h after scratching. The subtracted difference was calculated as the wound healing area.

### 2.8. Boyden Chamber Assay

8 µm-pore Corning^®^ FluoroBlok™ Cell Culture Inserts (Corning, Corning, NY, USA) were placed in a 24-well plate, and the upper and lower chambers were filled with 1 ng/mL fibronectin (Corning) and incubated for 2 h at 37 °C. After washing, the 24-well plate was filled with 700 µL of the six different culture media (*n* = 3), as described above, and HUVEC suspensions were placed in the upper chambers at a density of 5 × 10^4^ cells per well in 500 µL serum-free DMEM. After incubation for 12 h, migrated HUVECs on the lower surface of the chamber were fixed with 3% paraformaldehyde and stained with Alexa Fluor 488 Phalloidin (Thermo Fisher Scientific; Waltham, MA, USA) and 4′,6-diamidino-2′-phenylindole dihydrochloride (DAPI; Merck, Tokyo, Japan). Images were obtained from seven fields of each chamber by the DP70 camera under a fluorescence microscope (BX51; Olympus Optical, Tokyo, Japan), and migrated cells were counted using ImageJ software.

### 2.9. Matrigel Plug Assay

A total of 400 µL Matrigel^®^ Matrix Basement Membrane (Corning) was combined with 100 µL of the five different media (DMEM, DMEM with 19.1 ng/mL VEGF, whole SHED-CM, the <6 kDa fraction, or >6 kDa fraction) or 100 µL of a SHED suspension (5 × 10^5^ cells/mL) and injected subcutaneously into the back of each 8-week-old male C57BL/6J mouse (*n* = 7). The injections were performed slowly to allow the Matrigel to polymerize and form a jelly-like implant under the skin. Ten days later, the mice were sacrificed and the Matrigel plugs were removed with some surrounding tissue. The plugs were fixed in 4% paraformaldehyde for 4 h and then soaked in a 20% sucrose solution so that ice crystals did not form during freezing. Then, the plugs were embedded in Tissue-Tek^®^ O.C.T. compound (Sakura Finetek, Tokyo, Japan) and snap-frozen by immersion in liquid nitrogen-cooled isopentane. Sections (14 µm thick) were prepared and immunohistostained with an anti-CD31 antibody (BD Biosciences, Franklin Lakes, NJ, USA), which is expressed on the vascular endothelium, and DAPI. Effects of SHED-CM on cell migration were evaluated based on the number of endothelial cells which stained with both CD31 antibody and DAPI in each captured image.

### 2.10. Tube Formation Assay

96-well plates were coated with 50 µL Matrigel^®^ Matrix Basement Membrane and incubated for 30 min at 37 °C to promote gelling. HUVECs were seeded above the Matrigel^®^ matrix at a density of 4 × 10^4^ cells per well in 150 µL of the six different culture media as described above (*n* = 5). After 12 h of incubation, tube morphology was evaluated and images were captured by the DP70 camera. The total number of joints and total tube length per microscopic area were calculated using image analyzer software (Kurabo; Tokyo, Japan).

### 2.11. Aortic Ring Assay

Under a dissecting microscope, the descending thoracic aortas of 14-week-old male SD rats were isolated and cut into 2-mm sections. The aortic rings were embedded in Matrigel^®^ Matrix Basement Membrane and incubated in the five different media (DMEM, DMEM with 19.1 ng/mL VEGF, whole SHED-CM, the <6 kDa fraction, or >6 kDa fraction) (*n* = 3). The culture media were exchanged every 3 days. After 14 days, outgrowth of neovessels from the aortic ring was observed under a phase-contrast microscope (Keyence Corporation, Osaka, Japan). Images were obtained from seven random fields of each aorta. The total length of neovessels from the aorta per microscopic area was calculated with ImageJ software.

### 2.12. Statistical Analysis

Data are expressed as the mean ± standard deviation. Statistical analyses were performed using one-way analysis of variance with the Bonferroni test for post-hoc comparisons. Differences were considered significant at *P* < 0.05. All statistical analyses were carried out using EZR software (Saitama Medical Center, Jichi Medical University, Saitama, Japan) [31], which is a graphical user interface for R (The R Foundation for Statistical Computing, Vienna, Austria).

## 3. Results

When evaluating the effects of SHED-CM treatment, serum-free DMEM served as the control. VEGF was examined as a positive control because it is well recognized as an angiogenic factor. We also separated SHED-CM into less or more than 6 kDa fractions and examined which fraction had a beneficial effect on angiogenesis because only secreted factors of less than 6 kDa in SHED-CM promoted neurite outgrowth of mouse dorsal root ganglion neurons in our previous study [20]. Furthermore, we examined the effects of exosomes that were reported to be associated with angiogenesis [29] and found to be secreted from SHED in our previous study [20].

### 3.1. Effect of SHED-CM on Cell Viability of HUVECs

MTT assays were performed to analyze the effects of SHED-CM on cell viability of HUVECs. Incubation with whole SHED-CM significantly increased cell viability in a time-dependent manner compared with the control (relative ratios to the control after 24 and 48 h of incubation were 1.19 ± 0.06-fold and 1.61 ± 0.26-fold, respectively; Figure 1A,B). Similarly, the >6 kDa fraction increased cell viability after both 24 and 48 h of incubation (relative ratios to the control after 24 and 48 h of incubation were 1.26 ± 0.05-fold and 1.49 ± 0.18-fold, respectively; Figure 1A,B). The effects of whole SHED-CM and the >6 kDa fraction were less than those of VEGF after 48 h of incubation (Figure 1B). The <6 kDa fraction did not increase cell viability after 24 and 48 h of incubation (1.05 ± 0.03-fold and 0.82 ± 0.12-fold, respectively; Figure 1A,B). DMEM with exosomes increased cell viability only after 24 h, but not 48 h, of incubation (1.15 ± 0.03-fold and 1.22 ± 0.13-fold; Figure 1A,B). These data indicated that the >6 kDa fraction played an essential role in the effect of SHED-CM on promoting cell viability, but the effects were less than those of VEGF.

### 3.2. Effect of SHED-CM on Cell Migration of HUVECs

Wound healing and Boyden chamber assays were performed to analyze the effect of SHED-CM on cell migration of HUVECs. In the wound healing assay, SHED-CM morphometrically promoted cell migration of HUVECs (Figure 2A). Whole SHED-CM and especially the >6 kDa fraction significantly increased the distance of cell migration compared with the control (distances in response to whole SHED-CM, the >6 kDa fraction, and control were 262.0 ± 54.4 µm, 421.5 ± 55.0 µm, and 82.0 ± 37.8 µm, respectively; Figure 2B). Consistent with the results of the migrated distance, whole SHED-CM and especially the >6 kDa fraction increased the area covered by migrated cells compared with the control (areas in response to whole SHED-CM, the >6 kDa fraction, and control were 38.9 ± 16.6 × 10^4^ µm^2^, 80.2 ± 7.8 × 10^4^ µm^2^, and 14.1 ± 8.8 × 10^4^ µm^2^, respectively; Figure 2C). Moreover, both migration distance and wound healing area have shown significant differences between SHED-CM and the >6 kDa fraction. Meanwhile, the <6 kDa fraction and exosomes did not promote cell migratory activity (Figure 2B,C).

In the Boyden chamber assay, SHED-CM increased the number of migrated cells per field indicating HUVECs’ migratory activity (Figure 3). Whole SHED-CM and especially the >6 kDa fraction increased the number of migrated cells compared with the control (numbers of cells in response to whole SHED-CM, the >6 kDa fraction, and control were 99.6 ± 48.2 cells/field, 127.4 ± 60.7 cells/field, and 11.7 ± 8.4 cells/field, respectively; Figure 3B). Although the <6 kDa fraction and exosomes increased the number of migrated cells at the same degree as VEGF (40.0 ± 15.4 cells/field, 39.6 ± 21.4 cells/field and 48.5 ± 10.5 cells/field, respectively), the effect was much smaller than that of whole SHED-CM (Figure 3B).

The results of both wound healing and Boyden chamber assays suggested that the >6 kDa fraction played an essential role in the cell migration of HUVECs induced by SHED-CM and promoted cell migration comparably or more than VEGF.

### 3.3. In Vivo Effects of SHED-CM on Endothelial Cell Migration

Mouse Matrigel plug assays were performed to assess the effect of SHED-CM on endothelial cell migration in in vivo. Endothelial cell infiltration into the plugs was assessed by immunohistochemical analysis of CD31-positive cells in accordance with a previous report [32]. Morphometrically, SHED-CM promoted endothelial cell migration into the plugs (Figure 4A). Whole SHED-CM, the <6 kDa fraction, and >6 kDa fraction increased the number of CD31-positive cells recruited into the Matrigel plug compared with the control (numbers of CD31-positive cells in response to whole SHED-CM, the <6 kDa fraction, >6 kDa fraction, control, and VEGF were 94.3 ± 19.3 cells/field, 75.3 ± 25.7 cells/field, 103.7 ± 20.3 cells/field, 19.3 ± 4.5 cells/field, and 126.9 ± 61.3 cells/field, respectively; Figure 4B). The effect of whole SHED-CM, the <6 kDa fraction, and >6 kDa fraction on cell migration was comparable with that of VEGF, and there was no significant difference between VEGF and these three treatments (Figure 4B). These data suggested that both less and more than 6 kDa fractions played an important role in promoting cell migration by SHED-CM.

Moreover, to examine the effect of transplantation of SHED, we performed this assay with the plug containing SHED suspension (5 × 10^5^ cells/mL). The number of migrated endothelial cells was significantly higher in the plug containing SHED than that in the control (143.7 ± 24.6 cells/field). The effect of SHED was equal to that of SHED-CM and VEGF (Figure 4B), which suggested that the effect of transplanted SHED was mediated in a paracrine manner by secreted factors in SHED-CM.

### 3.4. Effect of SHED-CM on Tube Formation of HUVECs

Tube formation assays were performed to assess the effect of SHED-CM on tube formation of HUVECs. Morphologically, incubation in DMEM did not form complex tubular structures of HUVECs, whereas incubation in SHED-CM formed tubules and tubular rings (Figure 5A). The total tube length per field was extended for HUVECs incubated in whole SHED-CM, the <6 kDa fraction, and >6 kDa fraction compared with the control. The effect of SHED-CM was greater than that of VEGF (total tube lengths in response to whole SHED-CM, the <6 kDa fraction, >6 kDa fraction, control, and VEGF were 22.259.8 ± 1.104.8 pixels/field, 24.858.0 ± 1.649.0 pixels/field, 28.214.2 ± 854.3 pixels/field, 12.210.2 ± 1.001.2 pixels/field, and 16.122.2 ± 1.427.2 pixels/field, respectively; Figure 5B). Consistent with the results of the total tube length, the number of joints, which indicates the formation of HUVEC tubular networks, was also increased for HUVECs incubated in whole SHED-CM, the <6 kDa fraction, and >6 kDa fraction compared with the control, and the effect of SHED-CM was greater than that of VEGF (numbers of joints in response to whole SHED-CM, the <6 kDa fraction, >6 kDa fraction, control, and VEGF were 313.0 ± 32.5 counts/field, 372.4 ± 38.2 counts/field, 429.4 ± 29.0 counts/field, 165.2 ± 16.1 counts/field, and 232.2 ± 23.8 counts/field, respectively; Figure 5C). Moreover, the effect of both the <6 kDa and >6 kDa fraction were greater than that of SHED-CM. Exosomes promoted tube formation of HUVECs at the same degree as VEGF, but the effect was less than that of SHED-CM (Figure 5B,C).

These data suggested that the effect of SHED-CM was more than that of VEGF, and both less and more than 6 kDa fractions played an important role in promoting tube formation of HUVECs by SHED-CM.

### 3.5. Ex Vivo Effect of SHED-CM on Neovascularization

To verify the effect of SHED-CM on neovascularization, we performed rat aortic ring assays in ex vivo. Morphometrically, SHED-CM promoted neovessel sprouting from aortic rings of male Sprague-Dawley rats (Figure 6A). Whole SHED-CM, the <6 kDa fraction, and >6 kDa significantly extended the total neovessel lengths per field compared with the control, and the effect of SHED-CM was the same as that of VEGF (total neovessel lengths in response to whole SHED-CM, the <6 kDa fraction, >6 kDa fraction, control, and VEGF were 40.1 ± 15.7 pixels/field, 26.2 ± 8.1 pixels/field, 49.6 ± 13.4 pixels/field, 11.9 ± 4.8 pixels/field, and 32.5 ± 11.8 pixels/field, respectively; Figure 6B). The results of the aortic ring assay suggested that both less and more than 6 kDa fractions played an important role in promoting neovascularization by SHED-CM.

## 4. Discussion

Our recent study showed that SHED-CM prevented the progression of DPN by increasing the capillary number to muscle fiber ratio and capillary blood flow in diabetic mice [20]. In this study, we demonstrated that SHED-CM directly promoted angiogenesis through activation of endothelial cells. Therefore, we believed that SHED-CM might play an important role in treatment of diabetic induced vascular diseases.

Formation of new vessels involves two physiological processes, namely vasculogenesis and angiogenesis. Vasculogenesis is the developmental process of new blood vessel formation in the embryo, whereas, angiogenesis is defined as the outgrowth process of new blood vessels from pre-existing vasculature after birth [33]. Angiogenesis is highly organized by a sequence of cellular events. First, a cytokine gradient triggered by tissue hypoxia, tumorigenesis, or tissue injury induces destabilization of a vessel by digestion of the basement membrane. Then, angiogenesis is followed by sprouting, proliferation, and migration phases of endothelial cells to elongate a new vessel. Finally, a reorganization phase forms three-dimensional vessel structures through the fusion of new blood vessels [7]. In this study, the MTT assay showed the effect of SHED-CM on the cell viability. Wound healing, Boyden chamber, and Matrigel plug assays showed the effect of SHED-CM on the migration phase. The tube formation assay showed the effect of SHED-CM on the reorganization phase. Furthermore, the rat aortic ring assay reflects all phases of primary endothelial cells in angiogenesis. SHED-CM enhanced all endpoints of these assays, which indicates that SHED-CM directly promoted angiogenesis of endothelial cells throughout all phases. The process of angiogenesis is also tightly regulated by the balance between various pro- and anti-angiogenic factors [7]. In this study, 19.1 ng/mL VEGF was applied as a positive control and the concentration of VEGF in SHED-CM was approximately 1.4 ng/mL [20], which was very low compared with that of the positive control. However, SHED-CM promoted angiogenesis similarly or more than the positive control. Moreover, we previously reported that SHED-CM contains several angiogenic factors, such as FGF2, NGF, and BDNF [20,34,35,36], and a very recent study also demonstrated that SHED-CM contained higher levels of angiogenic factors than bone marrow-derived MSC-CM [24], although the concentrations of these factors in SHED-CM were not sufficient for bioactivity. Therefore, the cocktail of the angiogenic factors or unknown secreted factors might also be involved in the promotion of angiogenesis. In addition, the >6 kDa fraction, which contained the angiogenic factors such as VEGF, FGF2, NGF, and BDNF, showed greater effects than whole SHED-CM in wound healing assay, which implies that the <6 kDa fraction might contain angiogenic inhibitors. Various anti-angiogenesis factors have been identified such as angiostatin, endostatin, vasostatin, PF4, CXCL10 (IP-10), TIMP2, IFN-α1, IFN-β, IFN-γ, IL-4, IL-12, and IL-18 [37]. However, the molecular weights of all of these reported anti-angiogenic factors is more than 6 kDa, and it is still completely unknown which anti-angiogenic factors SHED-CM contains in the <6 kDa fraction. Meanwhile, in the tube formation assay, the effect of the <6 kDa fraction as well as >6 kDa fraction was significantly greater than that of whole SHED-CM. The discrepancy between the results of the tube formation assay and the others could be attributed to the quantification method. More detailed morphometric analysis using another cell imaging analyzer, such as Optomax (Hollis, NH, USA) which can measure tube area/field, might give us new insights and a better agreement with the other data. Further investigation should be needed in the near future. Compared to SHED, MSCs have been more widely proven the therapeutic effects on angiogenesis, and a number of angiogenic stimulators and inhibitors have been identified in secreted factors from MSCs, such as MCP-1, IL-8, SDF-1, VEGF, HGF, and angiopoietin-1 [22,38,39]. However, the detailed mechanisms of molecular interaction are still unclear and these factors in MSC-CM have been considered to be a balanced cocktail that acts in concert to promote angiogenesis. Another possibility for angiogenesis induced by SHED-CM is inorganic trace elements in SHED-CM, which might be associated with the mechanism of angiogenesis [40,41,42]. Iron has been reported to attenuate angiogenesis in in vitro by prevention of HIF signaling that leads to suppression of VEGF expression [43], while calcium has been reported to increase endothelial cell proliferation by activation of endothelial nitric oxide synthase [40]. Meanwhile, we considered that exosome in SHED-CM might contributed to the beneficial effects on endothelial cells to promote angiogenesis. However, the effect of exosomes on angiogenesis in in vitro experiments was statistically significant but very negligible compared to that of conditioned medium and fractions of SHED. Thus, we did not examine the effect of exosome in in vivo Matrigel plug assay and ex vivo Aortic ring assay. Taken together, the molecular details of SHED-CM might be different from that of MSC-CM, but we considered that a balanced cocktail of SHED-CM might play an important role in angiogenesis as well. Further experiments should be performed to identify the exact mechanism and secreted factors from SHED for promotion of angiogenesis using proteomics approach. In addition, as with MSCs, SHED has been demonstrated to develop a senescence phenotype with increasing passage number [24]. Senescent cells secrete inflammatory cytokines, proteases, and other factors, which are indicated as senescence-associated secretory phenotype (SASP) [44,45]. SASP has been reported to promote tumor development and malignant phenotypes through the proliferation and invasiveness [46]. Furthermore, in cell-culture models, SASP induced migration of endothelial cells [47]. Therefore, we should examine the role of SASP from senescent SHED in angiogenesis in the near future.

No clinical research of secreted factors from stem cells has been reported for any kind of disease. However, in recent years, transplantation of stem cells has been examined for therapeutic clinical application. In early clinical research, transplantation of bone marrow-derived mononuclear cells (BM-MNCs) was the most investigated to prove the potential clinical utility for ischemic vascular diseases because the cells were considered to supply endothelial progenitor cells and secrete angiogenic factors [48]. The TACT (Therapeutic Angiogenesis using Cell Transplantation) study first demonstrated the efficacy and safety of autologous BM-MNC transplantation for critical limb ischemia (CLI) [49], but subsequent meta-analysis of 10 randomized controlled trials showed no significant differences in major amputation rates, survival, or amputation-free survival between patients treated with BM-MNCs or placebos [50]. In recent clinical studies, transplantation of MSCs has gained interest and is the most widely examined as a new strategy for vascular diseases in both humans and animal models [51]. According to ClinicalTrials.gov, to date (March 2020), 23 clinical trials were registered to investigate therapeutic effects of MSC transplantation on CLI. Intra-arterial injection of allogeneic bone marrow-derived MSCs (BM-MSCs) has improved visual analog scale scores, the ankle brachial pressure index, and transcutaneous oxygen pressure [52]. Similarly, intramuscular injection of allogeneic BM-MSCs also demonstrated improvement of both rest pain and ulcers [53]. Based on the safety and feasible in these clinical trials, MSCs are now considered as a potential therapeutic option for ischemic diseases. However, no clinical research has examined the therapeutic effects of transplanting dental pulp stem cells, but several animal studies have revealed the effects of transplantation on ischemic diseases. Transplantation of SHED has healed hindlimb ulcers mainly by anti-inflammatory actions and promotion of angiogenesis in rat diabetic models [21]. In rat cerebral ischemic injury models, intravenous transplantation of DPSCs has improved functional recovery and showed superior reduction of the infarct size compared with that of BM-MSCs [17]. The underlying therapeutic mechanism of such transplantation is thought to be mediated in a paracrine manner. Thus, these studies support our findings that SHED also have the potential to promote angiogenesis in a paracrine manner. The clinical application of secreted factors from stem cells is still in the animal experimental stage. BM-MSC-CM has improved blood perfusion in a mouse limb ischemia model by enhancement of endothelial cell migration and tube formation [54]. Both transplantation of umbilical cord-derived MSCs (UC-MSCs) and administration of UC-MSC-CM show beneficial effects on skin regeneration and wound healing through angiogenic paracrine effects in diabetic mice. Moreover, interestingly, administration of UC-MSC-CM is therapeutically better than transplantation of UC-MSCs [55]. In the present study, the effect of SHED-CM on endothelial cell migration was the same as that of SHED suspension in the Matrigel plug assay, which suggests that SHED-CM treatment will have the same therapeutic effect as autologous transplantation of SHED in future clinical applications.

In conclusion, secreted factors from SHED accelerate neovascularization by activating endothelial cells and can be expected to be applied clinically. SHED-CM treatment is a cell-free therapy that might provide fewer possible adverse effects, such as immune compatibility and tumorigenicity, compared with cell transplantation [56]. Furthermore, CM can be stored easily for a long period without toxic cryopreservative agents [56]. This study showed that the >6 kDa fraction of SHED-CM was sufficient for treatment of vascular diseases. Moreover, the fractionation is preferable to eliminate unnecessary factors for angiogenesis and minimize the unknown side effects. However, the <6 kDa fraction promoted neurite outgrowth in our previous study, and another report has been reported that administration of whole SHED-CM to diabetic mice improved pancreatic function through β-cell proliferation [30]. Based on these results, we believe that whole SHED-CM administration could be a promising comprehensive treatment for diabetes, including improvement of microvascular complications, macrovascular disease and β-cell function. However, SHED-CM might cause adverse effects associated with angiogenesis such as deterioration of diabetic retinopathy or tumor growth. Therefore, further studies should be performed to understand the safety and precise mechanism of angiogenesis for future clinical applications of SHED-CM.

## Figures and Tables

**Figure 1 cells-09-02385-f001:**
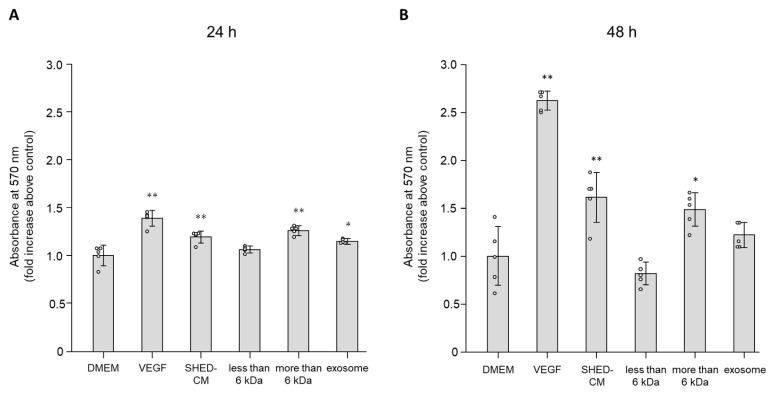
MTT assays were performed to assess cell viability of HUVECs. HUVECs were cultured in six different media (DMEM with 19.1 ng/mL VEGF, SHED-CM, the <6 kDa fraction, >6 kDa fraction, DMEM with exosomes, or DMEM as a control) for (**A**) 24 h and (**B**) 48 h (*n* = 5). Absorbance at 570 nm for each treatment group was normalized to the value in the DMEM treatment group. * *p* < 0.05 vs. DMEM, ** *p* < 0.01 vs. DMEM.

**Figure 2 cells-09-02385-f002:**
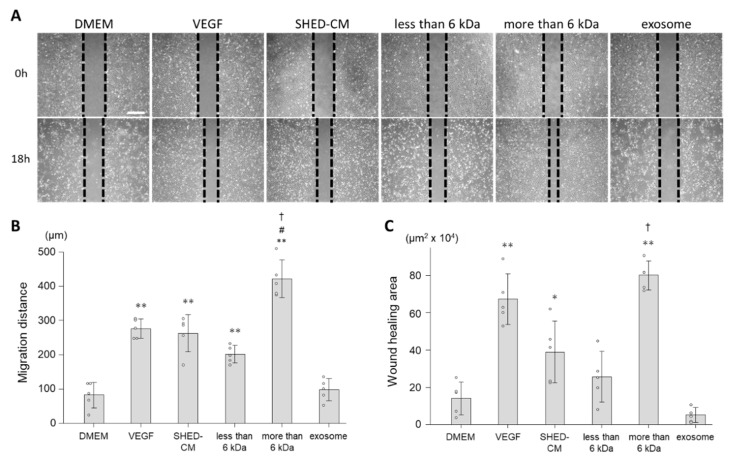
Wound healing assays were performed to assess HUVEC migration. (**A**) Representative images show HUVECs at 0 h (upper) and 18 h (lower) after scratching the cultures. HUVECs were cultured with VEGF (19.1 ng/mL), SHED-CM, the <6 kDa fraction, >6 kDa fraction, exosomes, or DMEM (*n* = 5). Scale bar = 500 µm. (**B**) Distances of cell migration. (**C**) Areas covered by migrated cells. * *p* < 0.05 vs. DMEM, ** *p* < 0.01 vs. DMEM, # *p* < 0.05 vs. VEGF, † *p* < 0.01 vs. SHED-CM.

**Figure 3 cells-09-02385-f003:**
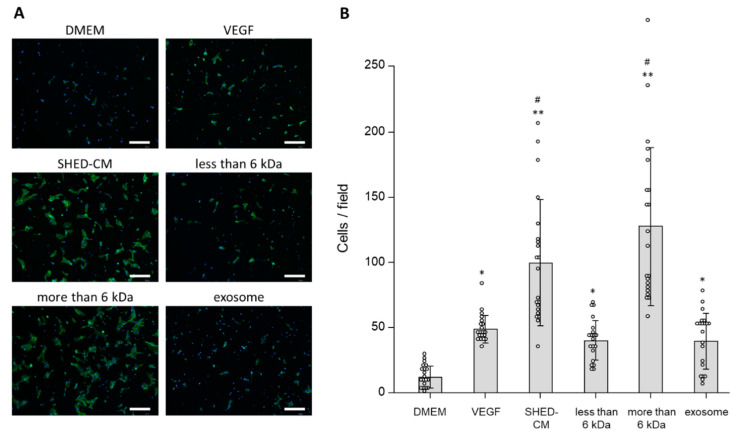
Boyden chamber assays were performed to assess HUVEC migration. (**A**) Representative images show HUVECs on the lower surface of the chamber. HUVECs were stained with phalloidin (green) and nuclei with 4′,6-diamidino-2′-phenylindole dihydrochloride (DAPI) (blue). HUVECs were incubated with VEGF (19.1 ng/mL), SHED-CM, the <6 kDa fraction, >6 kDa fraction, exosomes, or DMEM for 12 h (*n* = 3). Scale bar = 200 μm. (**B**) Numbers of cells that migrated to the lower surface of the chamber. * *p* < 0.05 vs. DMEM, ** *p* < 0.01 vs. DMEM, # *p* < 0.05 vs. VEGF.

**Figure 4 cells-09-02385-f004:**
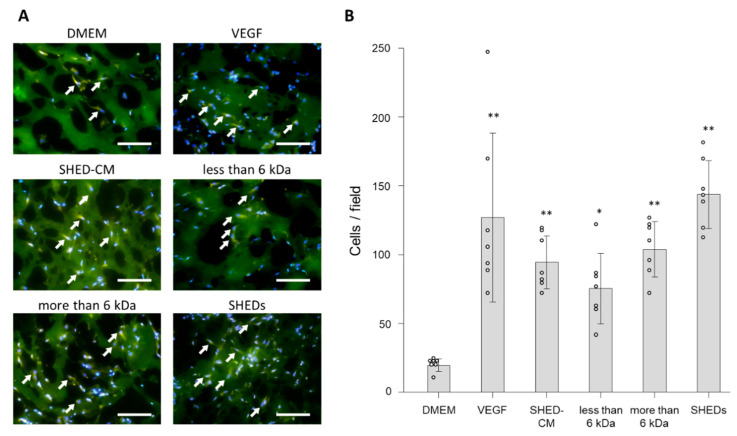
Matrigel plug assays were performed to assess angiogenesis. (**A**) Representative images show migrated endothelial cells (arrow) that were stained for the endothelial cell marker CD31 (green) and with DAPI (blue). Matrigel plugs containing VEGF (19.1 ng/mL), SHED-CM, the <6 kDa fraction, >6 kDa fraction, SHED, or DMEM were injected subcutaneously into male C57BL/6J mice (*n* = 7). Scale bar = 100 μm. (**B**) Numbers of endothelial cells that migrated into the Matrigel plug. * *p* < 0.05 vs. DMEM, ** *p* < 0.01 vs. DMEM.

**Figure 5 cells-09-02385-f005:**
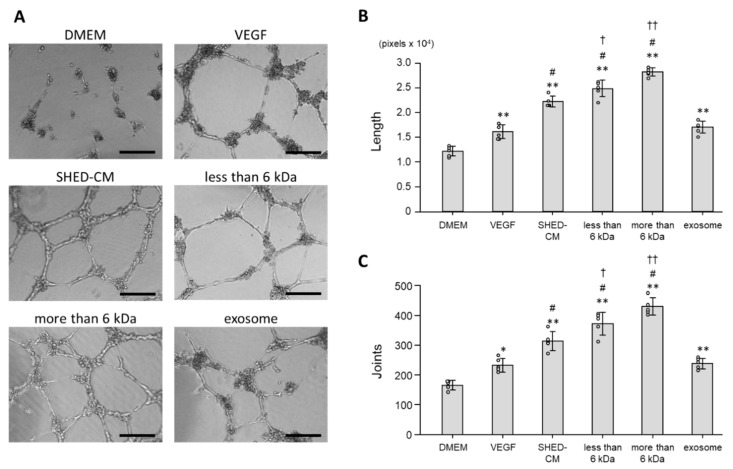
Tube formation assays were performed to assess HUVEC differentiation into tube-like structures. (**A**) Representative images show tube-like structures of HUVECs that were cultured for 12 h in the presence of VEGF (19.1 ng/mL), SHED-CM, the <6 kDa fraction, >6 kDa fraction, exosomes, or DMEM (*n* = 5). Scale bar = 200 μm. (**B**) Total tube lengths. (**C**) Numbers of joints. * *p* < 0.05 vs. DMEM, ** *p* < 0.01 vs. DMEM, # *p* < 0.05 vs. VEGF, † *p* < 0.05 vs. SHED-CM, †† *p* < 0.01 vs. SHED-CM.

**Figure 6 cells-09-02385-f006:**
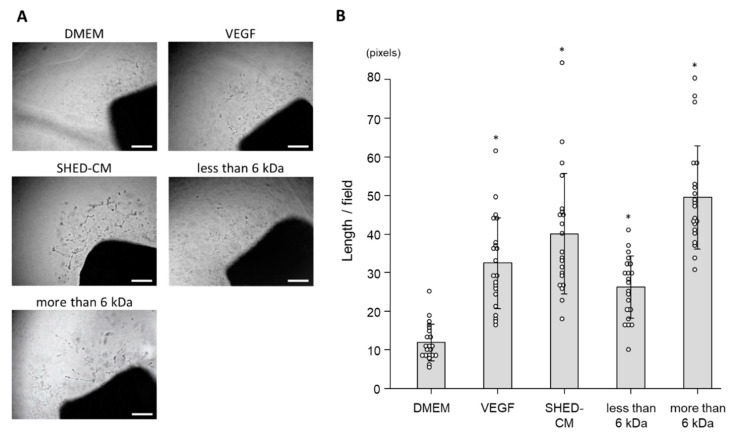
Aortic ring assays were performed to assess the effect of SHED-CM on angiogenesis. (**A**) Representative images show outgrowth of neovessels from the aortic ring of a male Sprague-Dawley rat. The aortic ring was embedded in Matrigel and incubated for 14 days with VEGF (19.1 ng/mL), SHED-CM, the <6 kDa fraction, >6 kDa fraction, or DMEM (*n* = 3). Scale bar = 500 μm. (**B**) Total lengths of neovessels. * *p* < 0.01 vs. DMEM.

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
