# Peer review of "Secreted Factors from Stem Cells of Human Exfoliated Deciduous Teeth Directly Activate Endothelial Cells to Promote All Processes of Angiogenesis"

_cells, 2020, doi:10.3390/cells9112385_

Round 1

Reviewer 1 Report

This study analyses the angiogenic potential of conditioned medium from stem cells of human exfoliated deciduous teeth. The conditioned medium has properties similar to those seen with exogenous VEGF. The authors conclude that such conditioned medium could play an important fole in future clinical application.

Whilst I cannot judge the physiological relevance of stem cells of human exfoliated deciduous teeth, it is clear that this study will offer very little to no interest to a broader readership.

The least that should be provided are molecular details of the composition of the conditioned medium, since by all criteria presented it is likely that one of the secreted factors is VEGF, the clinical potential of which has already been studied extensively. Without molecular details this study is prurely observational and offers no advance in the field and most importantly does not supprot the conclusion that the data presented bears translational impact.

Author Response

Reviewer 1

 We thank Reviewer #1 for their thoughtful comments and the encouragement of follow up experiments that emerge from this study.

 We quite agree that it is important to identify molecular details of SHED-CM. Our previous studies and others have demonstrated that engrafted SHED in various animal models of diseases promote significant recovery through endogenous tissue-repairing activities, but the precise molecular mechanisms are still unclear. As same as previous studies, we could not show the molecular details in this study. Meanwhile, compared to SHED, MSCs have been more widely proven the therapeutic effects on angiogenesis, and a number of angiogenic stimulators and inhibitors have been identified in secreted factors from MSCs, such as MCP-1, IL-8, SDF-1, VEGF, HGF, and angiopoietin-1 (Biomaterials. 2011 May;32(15):3689-99., Sci Rep. 2015; 5: 16295., Stem Cells. 2007 Oct;25(10):2648-59.). However, the detailed mechanisms of molecular interaction are still unclear and these factors in MSC-CM have been considered to be a balanced cocktail that acts in concert to promote angiogenesis.

 We previously identified a variety of secreted factors from SHED using cytokine arrays (J Neurosci. 2015 Feb 11;35(6):2452-64.), and also showed the concentrations of angiogenic factors such as VEGF and FGF2 in SHED-CM (J Diabetes Investig. 2020 Jan;11(1):28-38.). The concentration of VEGF in SHED-CM was 1.4 ng/mL, which was much lower than that in the positive control (19.1 ng/mL), suggesting that many other secreted factors from SHED might be involved in angiogenesis. In addition, in this study, the effects of exosomes derived from SHED-CM on angiogenesis were weak or negligible compared to those of SHED-CM, even though MSC-derived exosomes have been reported their efficacy on angiogenesis (Am J Physiol Heart Circ Physiol 2019;317:765-776.).

 Taken together, the molecular details of SHED-CM might be different from that of MSC-CM, but we considered that a balanced cocktail of SHED-CM might play an important role in angiogenesis as well. Further investigation should be continued to evaluate the detailed mechanisms using proteomics approach. We have discussed this in the text (page 11; lines 364-369, 374-382).

Reviewer 2 Report

This is an interesting article that looks to explore the clinical utility of angiogenic factors secreted from deciduous teeth and the authors should be commended for the work that has gone into this manuscript.

However, I do have a few concerns:

  1. The authors have used the MTT assay as a measure of cell proliferation as well as viability.  The MTT assay does not directly measure cell proliferation. It measures the metabolism of viable cells. A different assay should be used to measure proliferation.
  2. The relevance of using exosomes has not been addressed in the introduction to provide the context for why they are being used.
  3. CD31 antibody methods have not been provided.
  4. There are several different ECs and it is important to choose the right type of endothelial cells to resemble the context being studied. I question why the authors used HUVECs instead of using a cell line such as Human Pulmonary Artery endothelial cells or human cardiac microvascular endothelial cells.
  5. For the wound assay, a pipette tip was used. It is well known that this method is difficult to achieve reproducibility, but the authors have used image software to overcome this. Therefore, I would like to see just a little more info in the methods e.g. how many points were used, along the scratch, to measure area closed ( as it won’t be a uniform area) and how many times, over the 18h period were images taken.
  6. Figure 4/ Matrigel plug experiment shows that SHEDs were used in addition to conditioned medium and fractions but no exosomes. This is inconsistent with all the other experiments and reasoning is lacking.
  7. Lines 352 – 355 explains the reasoning for using 0.5nmol/l VEGF and what the concentration of VEGF is in the SHED-CM – this info should go into the methods to give context and justify concentrations used.

    8. The start of the discussion has a lot of waffle/re-amble that is not               needed and should be tailored to addressing the relevance of secreted         factors from stem cells to treat diabetic induced vascular disease.

Author Response

Reviewer 2

 We thank Reviewer 2 for appreciating the study and for their insightful comments. Our reply to each listed comment is below:

  1. We quite agree that MTT assay does not directly measure cell proliferation, because MTT assay just measures metabolic activity of the cells to convert water soluble MTT (3-(4,5-dimethylthiazol-2-yl)-2,5-diphenyltetrazolium bromide) compound to an insoluble formazan product. Thus, MTT assay reflects metabolic activities in the cells, and proliferation is considered as one of the activities. Meanwhile, it is reported that proliferation positively influences data of wound healing assay, in which secreted factors from SHED increased the area covered by migrated cells. In addition, we visually confirmed that the secreted factors from SHED increased the cell number in MTT assay and wound healing assay of this study, even though we did not examine anyquantitative analysis. Taken together, we considered that the secreted factors from SHED might have increased proliferation of HUVECs, but direct measurement, such as BrdU and CCK8 assay, should have be used to directly measure proliferation in this study. We removed the word “proliferation” in Result (page 5; lines 205-220) and explained the limitation for measurement of cell proliferation in this study in Discussion (page 10; lines 342-345).

  1. Thank you for pointing this out. We previously reported that conditioned medium from SHED contained enough exosomes to protectively effect on pancreatic beta-cells. MSC-derived exosomes also have been reported their efficacy in animal models of myocardial infarction, stroke, and hind-limb ischemia (Stem Cell Res 2013;10:301-312. Cell Physiol Biochem 2015;37:2415-2424. J Cereb Blood Flow Metab 2013;33:1711-1715. Stem Cell Res Ther 2015;6:10. Am J Physiol Heart Circ Physiol 2019;317:765-776.). Therefore, we considered that exosome in SHED-CM might contributed to the beneficial effects on endothelial cells to promote angiogenesis. We have now added this explanation in the text (page 2; lines 84-87).

  1. Thank you for the point. We have added the explanation of CD31 antibody in Materials and Methods of the text (page 4; lines 171-174).

  1. Thank you for the important point of HUVECs. Almost of all commercial endothelial cells, including HPAECs and HCMECs, are primary cultured cells derived from single donor endothelial cells. We considered that the cells derived from single donor were not appropriate for this study, because the angiogenic potential and VEGF responsiveness of the cells have been considered to be varied among lot numbers of cells. Meanwhile, HUVECs are derived from multiple donors, and considered to show steady response to VEGF. Numerous studies have used HUVEC as primary endothelial cells, and we also previously reported the study using HUVEC to evaluate angiogenic activity (Physiol Rep 2018;6(24):e13962.). Thus, we performed the experiments using HUVECs in this study.

  1. Thank you for the point. In the wound healing assay, five times experiments were performed for each condition. Images at the time of scratching were captured to measure the initial distance of wound at seven random sites. Then, the distance between the leading edge covered by cells 18 hours later were measured to evaluate the migration distance at same sites. In addition, the wound field was measured by ImageJ software at seven random sites at 0 and 18 hours after scratching. The subtracted difference was calculated as the wound healing area. We have now added this explanation in the text (page 4; lines 143-149).

  1. Thank you for pointing this out. The effect of exosomes on angiogenesis in in vitro experiments was statistically significant but very negligible compared to that of conditioned medium and fractions of SHED. Thus, we did not examine the effect of exosome in in vivo Matrigel plug assay and ex vivo Aortic ring assay. We discussed this in Discussion (page 11; lines 374-378).

  1. We agree with the Reviewer’s comment. We have now explained this in Materials and Methods of the text (page 3; lines 132-134).

  1. Thank you for raising the important points. In our previous study, SHED-CM prevented the progression of DPN by increasing the capillary number to muscle fiber ratio and capillary blood flow in the vasa nervorum of STZ induced diabetic mice (J Diabetes Investig 2020;11(1):28-38.). Therefore, we believed that secreted factors from SHED might directly promote angiogenesis through activation of endothelial cells and play an important role in treatment of diabetic induced vascular diseases. We have now simplified the start of discussion and explained the relevance of secreted factors from stem cells to treat diabetic induced vascular disease in Discussion (page 9; lines 323-327).

Reviewer 3 Report

In this paper the authors demonstrate that SHED-CM promotes angiogenesis through activation of endothelial cells, they performed several experiments analysing the effects of SHED-CM on each phase of the angiogenesis.

Before publication, the authors must clarify some issues or carry out further experiments to reinforce their conclusions.

One of the important points is to establish the state of the cells from which the conditional media is collected. In particular, it is necessary to check if in the cell culture there are senescent cells and in which percentage. This is crucial because it has been shown that senescent cells secrete inflammatory cytokines, proteases, and other factors, which are indicated as senescence-associated secretory phenotype (SASP) (PMID: 31235879; PMID: 27288264). The SASP released by damaged cells can promote surrounding cells to senesce. The SASP may also alter tissue and organ functions, thus contributing to individual aging (PMID: 24589226; PMID: 31412320; PMID: 16424011).

The authors show that the SHED-CM promotes cell viability and proliferation by MTT assay. This is a positive event to promote angiogenesis, but the authors must indicate in the discussion that in some cases the cellular secretome, in particular that of senescent cells, can stimulate cell transformation (PMID: 20078217; PMID: 26498687; PMID: 17409418).

In Materials and Methods, the authors must indicate the number of experimental replicates performed for each experiment.

The quality of Figure 4A needs to be improved.

In the discussion, the authors should indicate some future perspectives for identifying the molecules involved in the demonstrated effects on angiogenesis, for example carrying out an analysis of the SHED secretome to try to identify the most expressed molecules and which could be involved in promoting angiogenesis of endothelial cell.

Author Response

Reviewer 3

  1. One of the important points is to establish the state of the cells from which the conditional media is collected. In particular, it is necessary to check if in the cell culture there are senescent cells and in which percentage. This is crucial because it has been shown that senescent cells secrete inflammatory cytokines, proteases, and other factors, which are indicated as senescence-associated secretory phenotype (SASP) (PMID: 31235879; PMID: 27288264). The SASP released by damaged cells can promote surrounding cells to senesce. The SASP may also alter tissue and organ functions, thus contributing to individual aging (PMID: 24589226; PMID: 31412320; PMID: 16424011).

  1. The authors show that the SHED-CM promotes cell viability and proliferation by MTT assay. This is a positive event to promote angiogenesis, but the authors must indicate in the discussion that in some cases the cellular secretome, in particular that of senescent cells, can stimulate cell transformation (PMID: 20078217; PMID: 26498687; PMID: 17409418).

  1. In Materials and Methods, the authors must indicate the number of experimental replicates performed for each experiment.

  1. The quality of Figure 4A needs to be improved.

  1. In the discussion, the authors should indicate some future perspectives for identifying the molecules involved in the demonstrated effects on angiogenesis, for example carrying out an analysis of the SHED secretome to try to identify the most expressed molecules and which could be involved in promoting angiogenesis of endothelial cell.

 We thank Reviewer #3 for their appreciation of this study and insightful comments. Our reply to each listed comment is below:

Answer 1;

 Thank you for raising an important point that SASP from senescent SHED might promote surrounding cells to senesce. To the best of our knowledge, there is no report examined the role and function of senescent SHED. Meanwhile, there have been many reports on senescent MSCs, of which SASP promote surrounding cells to senesce. It is important to understand the biological features of senescent cells and the molecular mechanisms of SASP (Front Cell Dev Biol 2020;8:364.). Therefore, we should examine the role of SASP from senescent SHED in angiogenesis for future clinical applications of SHED-CM.

Answer 2;

 We totally agreed that cellular secretome of senescent cells can stimulate cell transformation. Secretome from senescent cells are reported to promote tumor development and malignant phenotypes through the proliferation and invasiveness (Annu Rev Pathol 2010;5:99-118.). Therefore, we should consider the effect of SASP from senescent SHED on HUVECs viability and proliferation in MTT assay. In addition, in cell-culture models, SASP induced migration of endothelial cells (J Biol Chem 2006;281(40):29568-74.). Further investigation should be needed to assess senescence level of SHED using acid beta-galactosidase assay. We have discussed this in the text (page 11; lines 382-386).

Answer 3;

 Thank you for the point. We have added the number of experimental replicates in the text.

Answer 4;

 Thank you for the important point. We improved the quality of Figure 4A with increasing pixel intensity and image contrast in accordance with the instruction of this journal.

Answer 5;

 We quite agree that it is important to identify molecular details of SHED-CM, which has also been addressed in comment of Reviewer 1.

 Our previous studies and others have demonstrated that engrafted SHED in various animal models of diseases promote significant recovery through endogenous tissue-repairing activities, but the precise molecular mechanisms are still unclear. As same as previous studies, we could not show the molecular details in this study. Meanwhile, compared to SHED, MSCs have been more widely proven the therapeutic effects on angiogenesis, and a number of angiogenic stimulators and inhibitors have been identified in secreted factors from MSCs, such as MCP-1, IL-8, SDF-1, VEGF, HGF, and angiopoietin-1 (Biomaterials. 2011 May;32(15):3689-99., Sci Rep. 2015; 5: 16295., Stem Cells. 2007 Oct;25(10):2648-59.). However, the detailed mechanisms of molecular interaction are still unclear and these factors in MSC-CM have been considered to be a balanced cocktail that acts in concert to promote angiogenesis.

We previously identified a variety of secreted factors from SHED using cytokine arrays (J Neurosci. 2015 Feb 11;35(6):2452-64.), and also showed the concentrations of angiogenic factors such as VEGF and FGF2 in SHED-CM (J Diabetes Investig. 2020 Jan;11(1):28-38.). The concentration of VEGF in SHED-CM was 1.4 ng/mL, which was much lower than that in the positive control (19.1 ng/mL), suggesting that many other secreted factors from SHED might be involved in angiogenesis. In addition, in this study, the effects of exosomes derived from SHED-CM on angiogenesis were weak or negligible compared to those of SHED-CM, even though MSC-derived exosomes have been reported their efficacy on angiogenesis (Am J Physiol Heart Circ Physiol 2019;317:765-776.).

Taken together, the molecular details of SHED-CM might be different from that of MSC-CM, but we considered that a balanced cocktail of SHED-CM might play an important role in angiogenesis as well. Further investigation should be continued to evaluate the detailed mechanisms using proteomics approach. We have discussed this in the text (page 11; lines 364-369, 374-382).

Round 2

Reviewer 1 Report

I am not satisfied by the answer to my main question regarding growth factor content of the SHED-CM. Whilst I can sympathise that the authors do not want to conduct elaborate direct studies of the compostion of the SHED-CM, they should consider the possiblity of growth factor (or glycon such as heprin) synergy before discounting the observations not related to VEGF. I would not be surprised to see a VEGF-neutralising agent completely abolishing the described effects.

However, given that the other reviewers are more enthusiastic I am prepared to give this study the benefit of the doubt provided the following is addressed adequately: 

Figure 1: Y axis needs to be labelled with a more specific description of what is depicted.

Figure 2: Why is the > 6 kDa fraction more potent than SHED-CM? Is the seeming difference significant (this pairwise comparison is important but not shown).

Figure 4: again more pairwise statistical comparisons are needed to understand if there are any differences with respect to the different angiogenic treatments. To me they look identical, which is rather curious since the in vitro analyses suggest that there are signficant differences in angiogenic potential.

Figure 4: Matrigel plug in vivo assays  are usually accompained by a somewhat more detailed analysis of the endothelial network that has invaded the matrigel plug.

Figure 5: Again the tube formation assays are at odds to previous studies. It is somewhat unusual to find formation of elaborate tube networks with cells that are not very migratory. Possibly a more detailed morphometric analysis of the data will reveal differences beyond those shown and in better agreement with the other data. This also needs to be discussed.

Figure 6: Same comment as for Figure 5 (and 4).

Discussion:

Lines 343+ : The statement "The MTT assay and Wound healing assay indirectly showed the effect of SHED-CM on the proliferation phase" is untenable, even with the caveat of "direct measurement, such as BrdU and CCK8 assay, should have been used to evaluate the proliferation phase." Neither MTT nor wound healing assays are a measure of proliferation as the former is influenced strongly by cell death and the latter is primarily a migration assay.

The following 3 sentences are also generalisations which are not accepted in the field of vasculare biology.

I am unclear why the authors chose to include the molarity of their VEGF treatment in all sections of the text, when the VEGF employed was bought by weight and is of undetermined molecular weight.

Lines 354+ : "Moreover, we previously reported that SHED-CM contains several angiogenic factors, such as FGF2, NGF, and BDNF [20,32-34], although the concentrations of these factors in SHED-CM were not sufficient for bioactivity. Therefore, unknown secreted factors might also be involved in the promotion of angiogenesis." The conclusion of the second sentence cannot be drawn as the authors have not tested a cocktail of these combined factors at these low concentrations. There is likely synergy between many of these.

"In addition, the >6 kDa fraction, which contained the angiogenic factors such as VEGF, FGF2, NGF, and BDNF, showed greater effects than whole SHED-CM in wound healing, Boyden chamber, aortic ring, and Matrigel plug assays, which implies that the <6 kDa fraction might contain angiogenic inhibitors." The first part of the statement is not demonstrated by adequate statistical comparisons. Furthermore, this could be easily demonstrated by mixing the two fractions at increasing concentrations.

Overall, the discussion speculates excessively when simple experiments could validate many of the ideas raised.

The authors should probably include the very recent publication 33006709, when assessing and discussing their own results.

Author Response

I am not satisfied by the answer to my main question regarding growth factor content of the SHED-CM. Whilst I can sympathise that the authors do not want to conduct elaborate direct studies of the compostion of the SHED-CM, they should consider the possiblity of growth factor (or glycon such as heprin) synergy before discounting the observations not related to VEGF. I would not be surprised to see a VEGF-neutralising agent completely abolishing the described effects.

However, given that the other reviewers are more enthusiastic I am prepared to give this study the benefit of the doubt provided the following is addressed adequately: 

  1. Figure 1: Y axis needs to be labelled with a more specific description of what is depicted.
  2. Figure 2: Why is the > 6 kDa fraction more potent than SHED-CM? Is the seeming difference significant (this pairwise comparison is important but not shown).
  3. Figure 4: again more pairwise statistical comparisons are needed to understand if there are any differences with respect to the different angiogenic treatments. To me they look identical, which is rather curious since the in vitro analyses suggest that there are signficant differences in angiogenic potential.
  4. Figure 4: Matrigel plug in vivo assays are usually accompained by a somewhat more detailed analysis of the endothelial network that has invaded the matrigel plug.
  5. Figure 5: Again the tube formation assays are at odds to previous studies. It is somewhat unusual to find formation of elaborate tube networks with cells that are not very migratory. Possibly a more detailed morphometric analysis of the data will reveal differences beyond those shown and in better agreement with the other data. This also needs to be discussed.
  6. Figure 6: Same comment as for Figure 5 (and 4).

Discussion:

  1. Lines 343+ : The statement "The MTT assay and Wound healing assay indirectly showed the effect of SHED-CM on the proliferation phase" is untenable, even with the caveat of "direct measurement, such as BrdU and CCK8 assay, should have been used to evaluate the proliferation phase." Neither MTT nor wound healing assays are a measure of proliferation as the former is influenced strongly by cell death and the latter is primarily a migration assay.

The following 3 sentences are also generalisations which are not accepted in the field of vasculare biology.

  1. I am unclear why the authors chose to include the molarity of their VEGF treatment in all sections of the text, when the VEGF employed was bought by weight and is of undetermined molecular weight.
  2. Lines 354+ : "Moreover, we previously reported that SHED-CM contains several angiogenic factors, such as FGF2, NGF, and BDNF [20,32-34], although the concentrations of these factors in SHED-CM were not sufficient for bioactivity. Therefore, unknown secreted factors might also be involved in the promotion of angiogenesis." The conclusion of the second sentence cannot be drawn as the authors have not tested a cocktail of these combined factors at these low concentrations. There is likely synergy between many of these.
  3. "In addition, the >6 kDa fraction, which contained the angiogenic factors such as VEGF, FGF2, NGF, and BDNF, showed greater effects than whole SHED-CM in wound healing, Boyden chamber, aortic ring, and Matrigel plug assays, which implies that the <6 kDa fraction might contain angiogenic inhibitors." The first part of the statement is not demonstrated by adequate statistical comparisons. Furthermore, this could be easily demonstrated by mixing the two fractions at increasing concentrations.

Overall, the discussion speculates excessively when simple experiments could validate many of the ideas raised.

  1. The authors should probably include the very recent publication 33006709, when assessing and discussing their own results.

 We thank the Reviewer for the insightful comments and the encouragement of follow up experiment. Our reply to each listed comment is below:

Answer 1;

 Thank you for pointing this out. Y axis is relative ratio to absorbance value of DMEM measured at 570 nm. We have now corrected this in the text and changed the Y axis description into “Relative ratio of absorbance at 570 nm over control” (page 6; lines 222-223).

Answer 2;

 Thank you for raising the important point. In the wound healing assay, both migration distance and wound healing area have shown significant differences between SHED-CM and the >6 kDa fraction (P < 0.01) using one-way analysis of variance with the Bonferroni test for post-hoc comparisons. We presume that the <6 kDa fraction might contain angiogenesis inhibitors, even though we did not conduct any experiment to identify the inhibitors. We have now added this explanation (page 6; lines 234-236) and discussed this in Discussion (page 10; lines 360-362), and shown these statistically significant differences in Figure 2.

Answer 3;

 Thank you for the point. In the Matrigel plug assay, whole SHED-CM, the <6 kDa fraction, and >6 kDa fraction increased endothelial cell migration significantly compared with DMEM as negative control. There was no significant difference between VEGF as positive control and all the three treatments by one-way analysis of variance with the Bonferroni test for post-hoc comparisons. We presumed that the partial inconsistency in the evaluation of endothelial cell migration between ex vivo Matrigel plug assay and other in vitro assays could be attributed to the quantification method, but further investigation should be performed in the near future. We have now added the explanation of pairwise statistical comparisons in the text (page 7; line 272).

Answer 4;

 Thank you for the important point. In general, in Matrigel plug assay, it is hard to quantify the neovascular network formed in the Matrigel. Many previous studies have quantified the amount of hemoglobin, the number of CD31-positive cells or the area of CD31-positive cells. Therefore, in this study, we counted the number of CD31-positive cells migrated into the Matrigel in accordance with a previous report (J Biol Chem. 2004;279(2):1304-9.), and quantified the effect of SHED-CM on angiogenesis. We have now added this explanation in the text (page 7; lines 264-265).

Answer 5;

 Thank you for pointing that. In this study, we investigated the tube formation of HUVECs with reference to previously reported method (J Biol Chem. 2004;279(2):1304-9.). We verified that HUVECs formed the tubular networks as the previous report and that positive controls significantly extended the total tube length and increased the number of joints compared with negative controls. Endothelial cells on Matrigel gradually form capillary-like tubular structures and connect with adjacent endothelial cells, then endothelial cells form two-dimensional branched structures that can lead to a meshed pseudo-capillary network. Thus, the tube formation assay is one of the most widely used in vitro methods that measures the cells' ability to form tubes, but not the ability to migrate. Taken together, we considered the evaluation of the total tube length and the number of joints adequate in this study.

 Meanwhile, we agree with the Reviewer’s line of thinking in regard to more detailed morphometric analysis. Only in the tube formation assay, the effect of the <6 kDa fraction was significantly greater than that of SHED-CM (P < 0.05), thus another cell imaging analyzer, such as Optomax (Hollis, NH, USA) which can measure tube area/field, might give us new insights and a better agreement with the other data, and further investigation should be needed in the near future. We discussed this in the text (page 11; lines 366-372), and shown these statistically significant differences in Figure 5.

Answer 6;

 Thank you for pointing that. In the Aortic ring assay we verified that neovessels sprouted out from aortic rings as previously reported (Blood. 2010;115(12):2520-32) and that positive controls significantly extended the total neovessel lengths compared with negative controls. Thus, we considered the evaluation adequate in this study.

Answer 7;

 We quite agree that neither MTT nor wound healing assays directly reflect proliferation potential. We have removed the pointed out sentence from the text and corrected this in the text (page 10; line 345).

Answer 8;

 We quite agree with the Reviewer’s comment. In this study, we chose to include the molarity of VEGF because previous study by us (J Diabetes Investig. 2020;11(1):28-38.) used DMEM with 0.5 nmol/L (19.1 ng/mL) VEGF as positive control. An experimentally effective concentration of VEGF has been reported to be from 10 to 20 ng/mL [28-30]. Recombinant Human VEGF165 is a 38.2 kDa, disulfide-linked homodimeric protein consisting of two 165 amino acid polypeptide chains, and the molarity of VEGF can be calculated from the molecular weight. We thank the Reviewer for raising the important point, and we will perform further studies using the solution concentration of VEGF as in many other studies.

Answer 9;

 We totally agree that the cocktail of the angiogenic factors at the low concentrations might have synergy potential for angiogenesis, even though the concentrations of each factor in SHED-CM is not individually sufficient for bioactivity. We have now explained the synergy potential in the text (page 10; lines 358-360). 

Answer 10;

 Thank you for pointing this out, and certainly >6 kDa fraction statistically showed greater effects than whole SHED-CM in wound healing assay. We have now removed the pointed out sentence from the text and corrected this in the text (page 10; lines 360-362), and added the result of statistical evaluation in the figure 2. Moreover, we thank the Reviewer for the encouragement of follow up experiment, and should make a detailed complementary follow study in the near future.

Answer 11;

 Thank you for informing the important and insightful paper for our study. We have now especially discussed about the senescence and secretome of mesenchymal stromal cells by citing this paper to improve our discussion (page 2; line 83, page 10; lines 356-357, page 11; lines 389-395).

Reviewer 3 Report

None

Author Response

Thank you once again for your consideration of our paper.

Round 3

Reviewer 1 Report

I am still not fully satisfied by the authors's response. The points (4-6) I made were meant to actually help elevate the conclusions drawn from this study. 

In any case they should consider the following small changes:

re Answer 1: "Absorbance at 570 nm (% control)" is usually used in similar datasets.

re Answer 8: the authors do not appear to understand the nature of VEGF. This heterodimeric ligand is secreted and thus glycosylated (potentially differentially depending on the secreting cell). Thus the MW derived from the polypeptide composition cannot be used to derive an exact molarity. This is why the vast majority of researchers operate by weight per volume. Even the provider of the VEGF used in this study refers to the MW as predicted and sells the polypeptide by weight.

Author Response

We appreciate the time and effort the Reviewer has dedicated to providing thoughtful and constructive comments. Our reply to each listed comment is below:

re Answer 1: "Absorbance at 570 nm (% control)" is usually used in similar datasets.

Thank you for pointing out. We have now changed the Y axis description into “Absorbance at 570 nm (fold increase above control)” (page 6; Figure 1).

re Answer 8: the authors do not appear to understand the nature of VEGF. This heterodimeric ligand is secreted and thus glycosylated (potentially differentially depending on the secreting cell). Thus the MW derived from the polypeptide composition cannot be used to derive an exact molarity. This is why the vast majority of researchers operate by weight per volume. Even the provider of the VEGF used in this study refers to the MW as predicted and sells the polypeptide by weight.

We quite agree with your thoughtful comments. In this study, 10 μg Recombinant Human VEGF165 was reconstituted at 100 μg/mL in sterile PBS, and then diluted to 19.1 ng/mL with an appropriate amount of DMEM. As we considered that the molecular weight of Recombinant Human VEGF165 was 38.2 kDa, we described 19.1 ng/mL VEGF as 0.5 nmol/L VEGF. In accordance with the Reviewer’s comment, we have now changed the description of “0.5 nmol/L” to “19.1 ng/mL”, and removed “0.5 nmol/L” from the text.

Again, thank you for giving us the opportunity to strengthen our manuscript with your valuable comments and queries. We should perform further investigation to clarify many important points you have raised in the near future.